# Identification of Key Aromatic Compounds in Basil (*Ocimum* L.) Using Sensory Evaluation, Metabolomics and Volatilomics Analysis

**DOI:** 10.3390/metabo13010085

**Published:** 2023-01-04

**Authors:** Pengmeng Du, Honglun Yuan, Yayu Chen, Haihong Zhou, Youjin Zhang, Menglan Huang, Yiding Jiangfang, Rongxiu Su, Qiyu Chen, Jun Lai, Lingliang Guan, Yuanhao Ding, Haiyan Hu, Jie Luo

**Affiliations:** 1College of Tropical Crops, Hainan University, Haikou 570288, China; 2Sanya Nanfan Research Institute of Hainan University, Hainan Yazhou Bay Seed Laboratory, Sanya 572025, China; 3Tropical Crops Genetic Resources Institute, Chinese Academy of Tropical Agricultural Sciences, Haikou 571101, China; 4National Key Laboratory of Crop Genetic Improvement and National Center of Plant Gene Research (Wuhan), Huazhong Agricultural University, Wuhan 430070, China

**Keywords:** *Ocimum basilicum*, basil, aroma, flavor, metabolome, volatilome

## Abstract

Basil (*Ocimum* L.) is widely used as a flavor ingredient, however research on basil flavor is limited. In the current study, nine basil species were selected, including *Ocimum basilicum* L.var. *pilosum* (Willd.) Benth., *Ocimum sanctum*, *Ocimum basilicum cinnamon*, *Ocimum gratissimum* var. suave, *Ocimum tashiroi*, *Ocimum basilicum*, *Ocimum americanum*, *Ocimum basilicum* ct linalool, and *Ocimum basilicum* var. *basilicum*, and their fragrance and flavor characteristics were assessed by sensory evaluation. The results indicated that *Ocimum basilicum* var. *basilicum* and *Ocimum gratissimum* var. *suave* have a strong clove smell and exhibited a piquant taste. Metabolomics and volatilomics analyses measured 100 nonvolatile metabolites and 134 volatiles. Differential analysis showed that eugenol, γ-terpinene, germacrene D and malic acid were among the most varied metabolites in basil species. Combined with sensory evaluation results, correlation analysis revealed that β-pinene and γ-cadinene contributed to the piquant smell, while eugenol and germacrene D contributed to the clove smell, and malic acid and L-(−)-arabitol contributed to the sweet flavor in basil. This study provided comprehensive flavor chemistry profiles of basil species and could be used as a guide for basil flavor improvement. The better understanding of objective sensory attributes and chemical composition of fresh basil could introduce the improved cultivars with preponderant traits, which is also in accordance with the various demands of breeders and growers, food producers, and consumers.

## 1. Introduction

The *Ocimum* genus, also known as basil, includes aromatic plants that are widely cultivated in the tropical and subtropical regions of Africa, Central South America and Asia. Basil and its indispensable oil have been widely used in the food, cosmetics, and pharmaceutical industries [1,2], and researchers have studied basils from genome, transcriptome and metabolome perspectives. Upadhyay et al. introduced the draft genome sequence of *O. tenuiflorum* (krishna subtype) and the transcriptome of two subtypes, krishna and rama Tulsi [3]. A large number of genes involved in the production of specific metabolites with medicinal value, such as apigenin, linalool and eugenol, have been identified [3]. In addition, transcripts of terpenoid synthetases in *O. sanctum* and transcripts of phenylpropanoid synthetase in *O. basilicum* were reported by comparing the transcriptomes of two *Ocimum* species [4]. More importantly, the volatile composition of basil and its essential oil, as well as their antioxidant, antibacterial and antifungal activities, have been extensively studied [2,5,6]. To date, more than ninety basil volatiles have been identified [7]. The prominent volatiles of basil include terpenoids (e.g., limonene, borneol, geranial and linalool) and benzene derivatives (e.g., estragole, methyl cinnamate and eugenol) [1,2,8]. Currently, researchers have focused on the study of volatiles with biological activities and/or antioxidant properties. For example, Lee et al. found that eugenol, thymol and carvacrol in basil exhibited strong antioxidant activities [8], while another report suggested that limonene, borneol, geranial and other terpenoids might contribute to the antidiabetic and antihypertensive properties of the basil essential oil [1]. However, basil is a popular flavor ingredient around the world, however research on basil flavor is limited. A recent study applied sensory evaluation to profile the aroma characteristics of seven basil varieties, and their volatilomes were assessed by untargeted volatilomics [9]. Dissimilarities in perceptual attributes among varieties were explored by linking sensory characteristics to volatile chemical components. This study effectively distinguished three basil characteristics from Italian varieties and revealed that substances such as 1,8-cineole and eugenol are related to aromas such as clove. Nonetheless, more work should be done to characterize basil flavor and screen for the contributory aromatic compounds.

Metabolomics studies combined with sensory analysis are an effective strategy to analyze the flavor chemistry and identify contributory compounds [10,11,12,13]. For instance, by using targeted metabolomics, researchers evaluated the flavor chemistry of tomato fruit and created a predictive and testable model of liking [10]. Based on this model, they found that citric acid, 3-methyl-1-butanol, 2-methylbutanal, and 1-octen-3-one were associated with flavor intensity, while 3-methyl-1-butanol and 2-methylbutanal were associated with sweetness, which provides new insights into tomato flavor. Similarly, Zhang et al. studied five mint genera at the sensory and metabolic levels and revealed that α-citral, menthofuran, isomenthone, menthol, carvone, and linalool are the key flavor compounds in mint [14]. These studies demonstrated that a metabolomics/volatilomics study combined with sensory analysis was an effective strategy to dissect the flavor composition and contributory compounds in plants. Therefore, we could apply a similar strategy to study basil flavor.

Currently, untargeted volatilomics is the most common technique used to profile plant volatilomes. The untargeted method emphasizes the detection of all detectable metabolites in the sample [15]. However, the method suffers from spectral convolution, low sensitivity, limited annotation coverage and poor reproducibility [15,16,17]. Recently, our group developed a novel widely targeted volatilomics (WTV) method [18]. Compared to the untargeted method, WTV addresses the data convolution problem, increases the coverage of annotated compounds, increases the sensitivity and enhances the reproducibility. Consequently, we could adopt WTV to obtain a more comprehensive basil volatilome profile.

In this study, nine basil species were selected, their volatilomes were assessed by the WTV method, and the nonvolatile metabolites were profiled by nontargeted metabolomics. Sensory analysis was carried out and combined with volatilome and metabolome data. We revealed the aromatic and metabolic differences within basil species and discovered the contributory compound to basil flavor, which provides a guide for its application in the food industry, such as edible essence, pesto and so on.

## 2. Materials and Methods

### 2.1. Plant Material

The leaves of all nine *Ocimum* species obtained from the Danzhou campus of Hainan University. These basils are grown and cultivated at the Hainan University base during spring–summer season. One to 1.5 cm width and 3–5 cm length leaves from eight months healthy plants (height around 40 to 60 cm) were harvested as samples. Leaf samples were collected randomly at the middle and upper parts from each plant, then cut into pieces and divided into three replications. For much bigger leaves, leaf samples from No. 4 *Ocimum gratissimum* var. *suave* were collected with leaves of 5 cm width and 7–8 cm length.

### 2.2. Chemicals

The hexane was acquired from Fisher Scientific (www.fishersci.com, Fair Lawn, NJ, USA, accessed on 1 January 2022). The methanol and N-Methyl-N-(Trimethylsilyl) Trifluoroacetamide (MSTFA) were purchased from Sigma-Aldrich (www.sigmaaldrich.com, St. Louis, MO, USA, accessed on 1 January 2022). Calcium chloride dihydrate, sodium chloride, pyridine and EDTA were obtained from Sinopharm Chemical Reagent Co., Ltd. (www.sinoreagent.com, Shanghai, China, accessed on 1 January 2022). The methoxyamine hydrochloride and n-alkanes (C8-C20) were purchased from Shanghai Aladdin Biochemical Technology Co., Ltd. (www.aladdin-e.com, Shanghai, China, accessed on 1 January 2022). All standards were purchased from Shanghai Aladdin Biochemical Technology Co., Ltd. (https://www.aladdin-e.com/, Shanghai, China, accessed on 1 January 2022) and Sigma-Aldrich (www.sigmaaldrich.com, St. Louis, MO, USA, accessed on 1 January 2022).

### 2.3. Sensory Analysis

A group of nine panelists (six males and three females) were recruited to evaluate nine basil species. The panel members were trained to evaluate the aroma of basil by orthonasal evaluation (smell) and evaluate flavor by retronasal evaluation (taste) with or without heat treatment. During retronasal evaluation, the panelists washed their mouth before they were served with new samples. Aroma was measured from the following attributes: Piquancy, spicy, numb-taste, minty, sweet, and clove, while flavor was measured from the following attributes: Piquancy, spicy, numb-taste, minty, sweet and bitter (Table 1). The intensities of taste and aroma sensations were evaluated from 1 (detectable) to 10 (very intensive). To evaluate taste score, 0.5 g samples from each plant were prepared for nine panelists, all scores were averaged across panelists (*n* = 9) and plotted on a radar chart.

### 2.4. Sample Preparation

At harvest, basil leaves were thoroughly rinsed with tap water and nanopure water to remove the residual soil or particles from the surface. Fresh basil leaves samples were flash frozen in liquid nitrogen. The frozen material was homogenized in a cryogenic mill. To perform volatilomics analysis, 0.1 g of the resulting powder was transferred into a 22-mL glass headspace vial, incubated for 10 min at 37 °C, and then 0.2 g of CaCl_2_·2H_2_O and 0.2 mL of a 100 mM EDTA-NaOH solution (pH 7.5) were added, gently mixed and sonicated for 5 min. Quality control (QC) samples were prepared by mixing aliquots of all samples. Samples were preheated for 10 min at 50 °C and extracted for 20 min at 50 °C. Three repetitions were performed for each basil species. To perform metabolomics analysis, 0.1 g sample was transferred to a 2 mL centrifuge tube. Then 1 mL of 70% cold methanol were added and vortexed. The sample was placed at 4 °C and extracted for 12 h. The sample was centrifuged at 12,000 g for 10 min at 4 °C. QC sample was prepared by mixing aliquots of all samples. A 200 μL aliquot of supernatant was transferred to a new 1.5 mL centrifuge tube for vacuum-drying at 4 °C. Afterward, 100 μL of 20 mg/mL methoxylamine hydrochloride in pyridine was added. The mixture was vortexed vigorously for 2 min and incubated at 37 °C for 60 min. One hundred microliters of MSTFA were then added, and the sample was derivatize at 60 °C for 180 min. The derivatized sample was filtered by a 0.22 μm filter before GC-MS analysis.

### 2.5. GC-MS Analysis

Metabolome and volatilome were profiled by gas chromatography (7890A GC, Agilent Technologies, Santa Clara, CA, USA) with an Agilent 7000D mass selective detector. HP-5 MS capillary column (30 m × 0.25 mm i.d, 0.25 μm film thickness; Agilent Technologies, Santa Clara, CA, USA) was used to separate compounds. To perform volatilomics analysis, the temperature program was as follows: The initial column temperature was 40 °C, held for 3 min, with a temperature increase of 2 °C/min temperature of 160 °C, and a temperature increase of 50 °C/min to a final temperature of 300 °C after reaching 160 °C, followed by a 3 min preservation at 300 °C. The injection temperature was 270 °C in splitless mode with a 0.75 mm i.d. inlet liner tube (Agilent Technologies, Santa Clara, CA, USA). The flow rate was He 1.0 mL/min (99.999%). Basil volatiles were first detected by full scan mode, then these signals were converted to multiple reaction monitoring (MRM) transitions and integrated into WTV MS2T library according to previous report [18]. A fiber with a usage count of ~70 was used to perform volatilomics analysis to ensure method reproducibility. The QC samples were injected at regular intervals (every 14 samples) throughout the analytical run. To perform metabolomics analysis, the temperature program was as follows: the initial oven temperature was 70 °C and held at 70 °C for 3 min, ramped to 300 °C at a rate of 10 °C/min, and finally held at 300 °C for 5 min. The injection volume was 1 μL with the injector temperature 270 °C in split 10:1 mode. The collision energy of full scan mode was 70 eV. The scanned range was 50–650, and the solvent delay time was set to 5.4 min. The QC samples were injected at regular intervals (every 14 samples) throughout the analytical run.

### 2.6. Compound Identification

C8-C20 alkane standard mix solution was measured using the same temperature program to calculate the retention index (RI). Signals were deconvoluted by MS-DIAL (Version 4.70) and identified by comparing the deconvoluted mass spectra and RI with those reported in the NIST library (Version 2.3) [19]. Commercially available standards were purchased and analyzed to confirm the identification results. The standards used in the study are shown in Appendix A (identification level A).

## 3. Results and Discussion

### 3.1. Sensory Evaluation

Nine basil species were selected for flavor analysis in this study, including *Ocimum basilicum* L. var. pilosum (Willd.) Benth, *Ocimum sanctum*, *Ocimum basilicum* cinnamon, *Ocimum gratissimum* var. suave, *Ocimum tashiroi*, *Ocimum basilicum*, *Ocimum americanum*, *Ocimum basilicum* ct linalool, and *Ocimum basilicum* var. basilicum (Figure 1). We first evaluated the fragrance and flavor of basil species. Volatiles are perceived in two ways: They can be sniffed through the nostrils (orthonasal olfaction), or when foods containing volatiles are chewed and swallowed, volatiles are forced up behind the palate into the nasal cavity posteriorly (retronasal olfaction) [10]. Therefore, we evaluated the basil fragrance by smell (orthonasal olfaction evaluation) and evaluated its flavor by taste (retronasal olfaction evaluation) with or without heat treatment (heat treatment was used to stimulate the cooking process). Fragrance and flavor were measured from several attributes, including overall intensity (Table 1). It should be noted that not everyone agreed on the ‘‘best’’-tasting basil; therefore, the sensory analysis rated overall intensity instead of overall liking. The results (Figure 2) showed that in orthonasal olfaction evaluation, most of the basil species showed a piquant smell. In particular, *Ocimum gratissimum* var. suave (No. 4) has a strong clove odor, while *Ocimum basilicum* L. var. pilosum (Willd.) Benth (No. 1) has the weakest overall fragrance. In retronasal olfaction evaluation without heat treatment, most basils exhibited a spicy taste, while *Ocimum basilicum* cinnamon (No. 3) had a characteristic sweet taste, and *Ocimum gratissimum* var. suave (No. 4) exhibited a piquant, minty, and numb taste. Again, *Ocimum basilicum* L. var. pilosum (Willd.) Benth (No. 1) has the weakest overall taste. Our results indicated that spicy is the dominant odor and taste in basil species, while *Ocimum gratissimum* var. suave (No. 4) has a unique clove smell, and *Ocimum basilicum* cinnamon (No. 3) has a characteristic sweet taste, which shows its high value in the food industry. In addition, we measured the same characteristics in retronasal olfaction evaluation with heat treatment and found that the results were similar but diminished. Therefore, the heat treatment rating was excluded in subsequent analysis to obtain a concise result.

### 3.2. Metabolomics Analysis

After evaluating the fragrance and flavor differences, we performed WTV and untargeted metabolomics to study the metabolic differences among basil species. A total of 134 volatile signals were measured, of which 94 could be identified. Meanwhile, 100 nonvolatile metabolite signals were detected, of which 86 could be identified. Volatiles included terpenes, benzene derivatives, esters, alcohols, ketones and aldehydes, while nonvolatile metabolites included organic acids, amino acids, sugars and fatty acids (Appendix A). Due to the high sensitivity and increased compound annotation of the WTV method, the amount of detectable and identified volatiles was higher than that of previous studies [20,21,22]. In accordance with previous reports, the majority of basil volatiles are terpenes, in which humulene, germacrene D, γ-cadinene, *trans*-α-bergamotene, and γ-cadinol are among the most abundant volatiles [23]. Moreover, the WTV method additionally detected high levels of eucalyptol and fenchol, which provide a more comprehensive basil volatilome profile. All plants synthesize a diverse array of terpenoids, and terpenoids comprise the largest, most chemically, structurally and functionally diverse class of chemicals in living organisms [24]. Terpenes are mostly isomers and are difficult to distinguish by mass spectra. In our MS2T library, most of the terpenes have the same/similar MRM transitions. To obtain an accurate result, (i) most of the terpenes were separated according to RI, and (ii) for terpenes sharing very similar RIs, e.g., β-copaene and β-gurjunene, these compounds were identified by authentic standards if possible, and during quantification, the integration of coeluting signals was manually adjusted. In nonvolatile metabolomic analysis, we found that galactinol, sucrose, rosmarinic acid, myo-inositol and catechollactate were the most abundant metabolites in basil. Similar to terpenes, some sugars are isomers and were identified by RI and authentic standards.

We then subjected the volatilome and metabolome data to principal component analysis (PCA) (Figure 3A,B). The first principal component (PC1) separated the basil species in both plots, and PC1 accounted for 27.4% and 27.9% of the total variance in volatilome and nonvolatile metabolome data, respectively, which indicated significant metabolic differences among basil species. In addition, *Ocimum gratissimum* var. *suave* basil (No. 4) was distinctly separated from other species in the PCA plot, which highlighted the consistency between our metabolome and sensory evaluation data. Then, we applied partial least squares discriminant analysis (PLS-DA) and calculated the variable importance in the projection (VIP) value to screen differential metabolites. The results revealed that in the volatilome data, eugenol, β-gurjunene, α-thujene, *trans*-β-ocimene, germacrene D, α-cadinene, cubenol, cubenene, β-calacorene, α-terpinene, γ-terpinene, α-calacorene, terpinen-4-ol, cadina-1(10),4-diene and cis-2-p-menthen-1-ol have the highest VIP value, while in the metabolome data, glycerol, maltose, D-(+)-xylose, d-galactose, dulcitol, D-ribose, glucose, D-mannitol, D-(−)-erythrose, erythritol, L-(−)-sorbose, malic acid and D-sorbitol have the highest VIP value. These compounds were responsible for the discrimination of basils species (Figure 3C,D). Specifically, eugenol, *trans*-β-ocimene and germacrene D were among the most varied volatiles within basil species, in which No. 4 had the highest compounds content with a level eight times higher than the average (Figure 4). Given that No. 4 has a strong clove smell, we assumed that these compounds might contribute to the clove smell in basil. In the metabolome analysis, malic acid levels varied greatly among basil species, and No. 3 had the highest level, which was three times higher than the average (Figure 4). Since No. 3 has a characteristic sweet taste, we hypothesized that malic acid might contribute to the sweet taste of basil [25,26]. Our results showed that terpenes, sugars and organic acids are significantly varied among basil species. To confirm their contributory role in basil aroma and flavor, we combined the metabolomics/volatilomics data and sensory evaluation results and carried out correlation analysis.

### 3.3. Identification of Contributory Aromatic Compounds in Basil

Traditionally, the major odorants are screened according to odor activity values, ignoring changes in the rate at which odor intensity grows beyond a threshold value. In addition, the chemical composition of a food does not in itself tell us whether the food will be liked [10]. Therefore, we conducted a correlation analysis between compound abundance and sensory evaluation rating to elucidate the flavor chemistry of basil (Figure 5). Three datasets were generated: (i) Correlations between the rating (overall intensity and each sensory attribute) of orthonasal olfaction evaluation and volatile abundance were calculated, which could screen the key aromatic volatiles [27], (ii) the correlation between the rating of retronasal olfaction evaluation (without heat treatment) and volatile abundance was calculated to identify key flavor volatiles, and (iii) correlations between the rating of retronasal olfaction evaluation and nonvolatile metabolite abundance were also calculated to identify key flavor metabolites. Rating of retronasal olfaction evaluation with heat treatment was excluded, as mentioned above.

Our sensory evaluation results indicated that the piquant and clove smells are important sensory attributes of basil smell, with spicy and sweet being important attributes for basil flavor. The correlation analysis results showed that β-pinene and γ-cadinene were positively correlated with the piquant smell rating (Figure 5), while eugenol, humulene, germacrene D and *trans*-β-ocimene were positively correlated with the clove smell (Figure 5). However, hydrocaffeic acid and ferulic acid are positively correlated with the spicy flavor (Figure 5), while malic acid and L-(−)-arabitol are positively correlated with the sweet flavor (Figure 5). To confirm the contributory roles of the aromatic compounds, commercially available standards were purchased, and the smell of diluted standard solutions was evaluated. The results indicated that β-pinene and γ-cadinene contributed to the piquant smell, while eugenol and germacrene D contributed to the clove smell in basil. These findings were in accordance with a previous study, in which researchers showed that methyl chavicol and eugenol contribute to anise and clove flavors [9]. However, the correlation analysis strategy and WTV method enabled us to identify β-pinene and γ-cadinene as novel aromatic compounds that contributed to the piquant flavor in basil. In addition, retronasal olfaction evaluation revealed the contributory roles of hydrocaffeic acid, ferulic acid, malic acid, and L-(−)-arabitol to spicy and sweet flavors. Our findings provide novel insight into basil flavor chemistry and could be used as a guide for basil flavor improvement.

An interesting result is that some volatiles, such as cis-anethole, epi-cubenol and α-bulnesene, are positively correlated with the sweet perception of basil (Appendix A). This result was similar to that of a previous report, in which researchers found that geraniol affects tomato sweetness [10]. We also found that eugenol, humulene, germacrene D, β-pinene and *trans*-β-ocimene contribute not only to the smell of piquancy but also to the taste of piquancy, and they also promote a numbing taste in basil. Volatiles play an important role in postnasal olfaction, and instances of volatile-induced tastes of sweet, sour, bitter, and salty have been observed [28]. Similarly, our findings suggested that increasing the content of certain volatiles is an alternative strategy to improve the flavor of fruits and vegetables.

Previous studies have shown that the flavor compounds identified in this study, such as eugenol, methyl-chavicol, linalool and nerol, have antioxidant, antibacterial and antifungal activities and have been widely used in the pharmaceutical industry [5,29,30,31,32]. In addition, terpinen-4-ol exhibited a depressant effect on the central nervous system [33,34]. Malic acid is involved in the important TCA cycle in living organisms and is an essential organic acid that can be used to treat liver disease, anemia, and immune deficiency [25,26]. Our study revealed that, in addition to their pharmaceutical value, these compounds also contribute to basil flavor.

## 4. Conclusions

This research comprehensively studied the differences among basil species by using sensory evaluation and chemometric technologies. The metabolome and volatilome results combined with sensory analysis revealed the contributory aroma compounds in basil. Specifically, β-pinene and γ-cadinene are responsible for the piquant smell, the piquant flavor, while eugenol, humulene, germacrene D and *trans*-β-ocimene contributed to the clove smell, and flavor in basil. This research provides new insight into the chemical composition of basil as well as guidance for basil flavor improvement. This study is valuable in increasing the chemical knowledge of basil aroma, provides theoretical guidance for basil breeding and quality control, and aids the effective utilization of basil resources and the development of innovative products with better flavor and higher added value.

## Figures and Tables

**Figure 1 metabolites-13-00085-f001:**
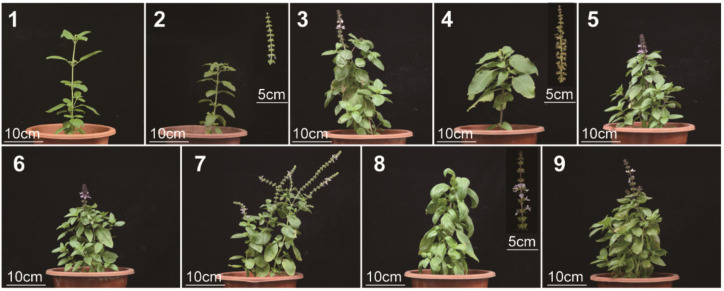
Phenotypes of different *Ocimum* species. Number 1–9: NO. 1 *Ocimum basilicum* L.var. *pilosum* (Willd.) Benth., NO. 2 *Ocimum sanctum*, NO. 3 *Ocimum basilicum cinnamon*, NO. 4 *Ocimum gratissimum* var. *suave*, NO. 5 *Ocimum tashiroi*, NO. 6 *Ocimum basilicum*, NO. 7 *Ocimum americanum*, NO. 8 *Ocimum basilicum* ct linalool, and NO. 9 *Ocimum basilicum* var. *basilicum*, respectively.

**Figure 2 metabolites-13-00085-f002:**
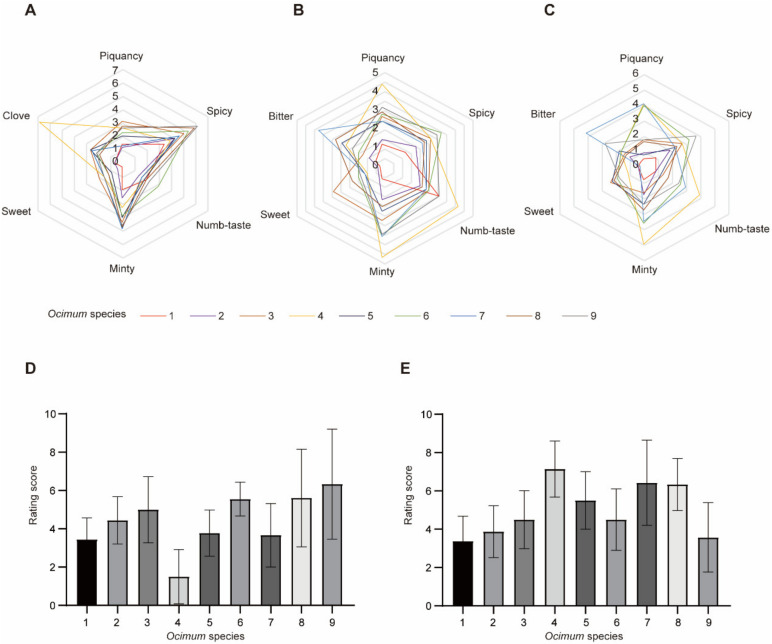
Sensory evaluation of different basil species. (**A**–**C**): Orthonasal evaluation, retronasal evaluation without heat treatment and retronasal evaluation with heat treatment, respectively. (**D**,**E**): Rating score of overall intensity of basils in orthonasal evaluation and retronasal evaluation without heat treatment, respectively. Data are represented as mean ± SD, each included nine replications.

**Figure 3 metabolites-13-00085-f003:**
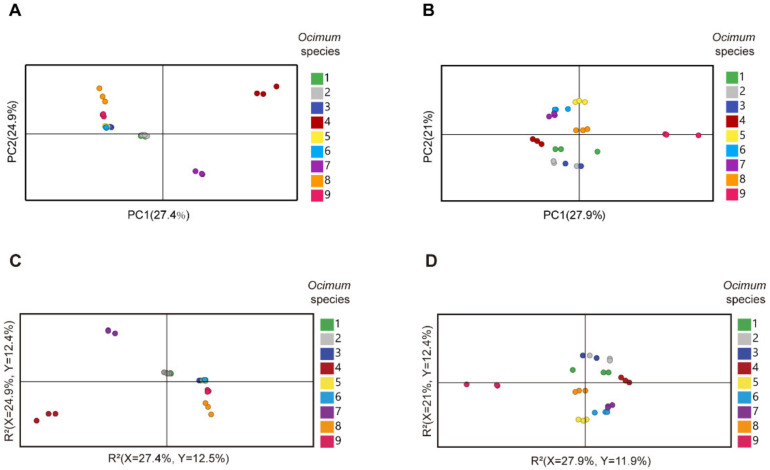
Multivariate analysis of metabolome and volatilome data of different basil species. (**A**): PCA (principal component analysis) plot of volatilome data. (**B**): PCA plot of metabolome data. (**C**): PLS-DA (partial least squares discriminant analysis) plot of volatilome data. (**D**): PLS-DA plot of metabolome data.

**Figure 4 metabolites-13-00085-f004:**
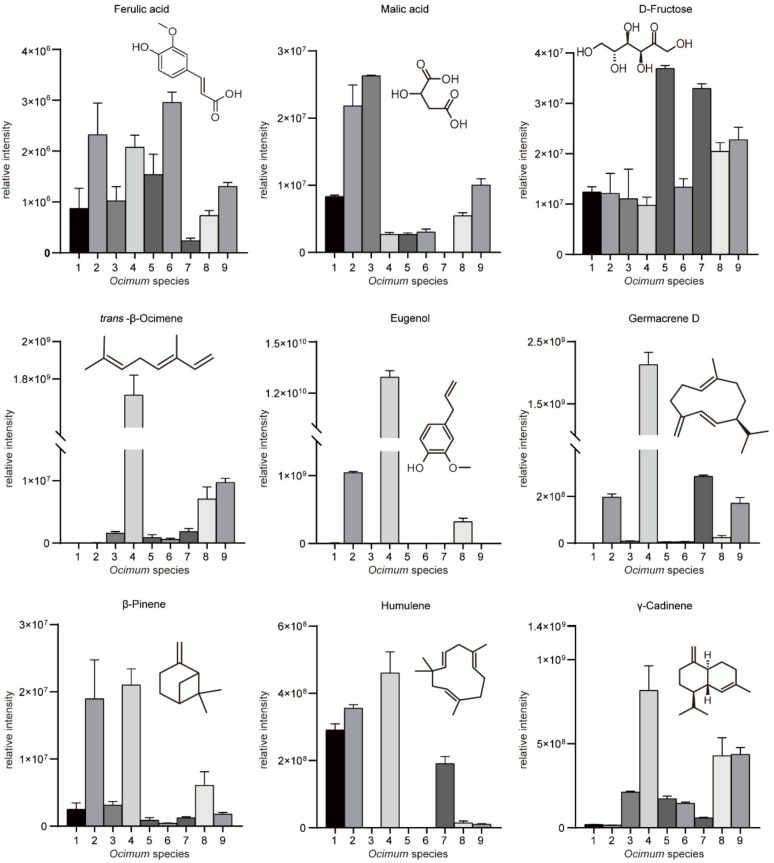
Relative intensities of key aromatic/flavor compounds in basil identified in this study.

**Figure 5 metabolites-13-00085-f005:**
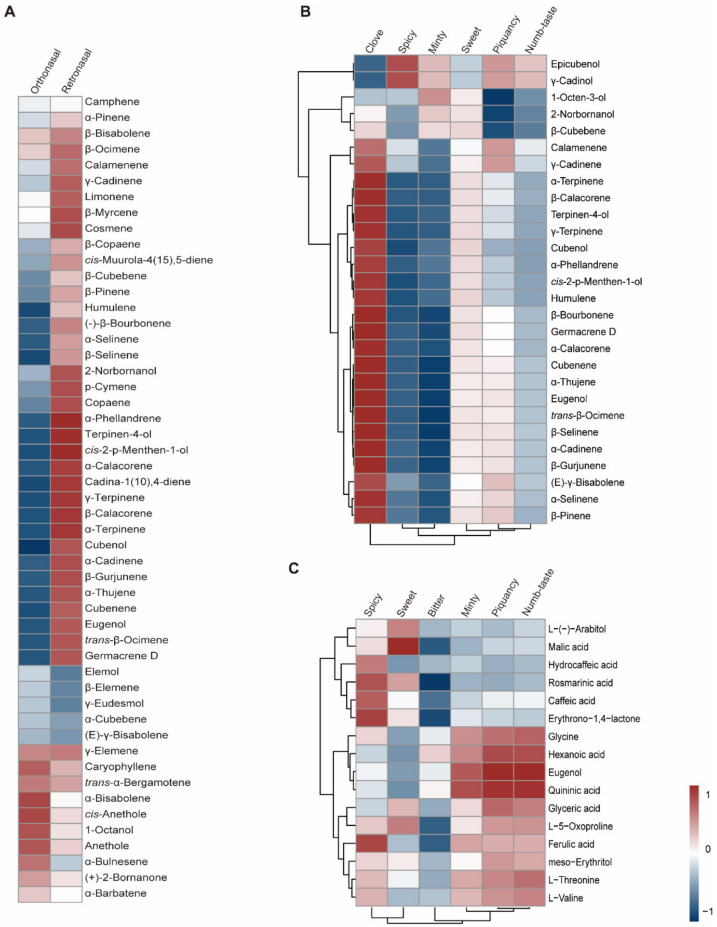
Correlation analysis between sensory evaluation results and metabolome/volatilome data. (**A**): Correlation analysis between overall intensity in orthonasal/retronasal evaluation and volatilome data. (**B**): Correlation analysis between different attributes rating in orthonasal evaluation and volatilome data. (**C**): Correlation analysis between different attributes rating in retronasal evaluation and metabolome data. Red and blue indicates positive and negative correlation, respectively.

**Table 1 metabolites-13-00085-t001:** Definitions of sensory attributes for basil.

Sensory Attribute	Definition
Piquancy	A sharp prickling sensation and a special burning sensation
Spicy	Test and smell like aniseed
Numb-taste	Test and smell of Sichuan pepper
Minty	Test and smell of menthol
Sweet	Elementary taste
Clove	Test and smell of clove
Bitter	Test and smell like bitter gourd

## Data Availability

Not applicable.

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
