# Peer review of "Identification of Key Aromatic Compounds in Basil (Ocimum L.) Using Sensory Evaluation, Metabolomics and Volatilomics Analysis"

_metabolites, 2023, doi:10.3390/metabo13010085_

Round 1

Reviewer 1 Report

The paper entitled “Identification of key aromatic compounds in basil (Ocimum L.) using sensory evaluation, metabolomics and volatilomics analysis” is focused on investigation of the volatile nonvolatile metabolites of the nine basil species. Authors present interesting studies that can be used for effective basil application in the food industry as well as for the selection of basil species with improved chemical and flavor properties.

The manuscript is well prepared, nicely organized and written. The paper needs however a minor revision. Please reexamine the paper taking into account the following:

1. Abstract: I recommend adding the names of the nine studied basil species to the Abstract.

2. Introduction, L41: Kindly check the compound “rosinic acid”. It is not typical for basil.

3. Introduction and references: Please check the reference # 18: the paper was published in 2022 (Honglun Yuan, Guangping Cao, Xiaodong Hou, Menglan Huang, Pengmeng Du, Tingting Tan, Youjin Zhang, Haihong Zhou, Xianqing Liu, Ling Liu, Yiding Jiangfang, Yufei Li, Zhenhuan Liu, Chuanying Fang, Liqing Zhao, Alisdair R. Fernie, Jie Luo. Development of a widely targeted volatilomics method for profiling volatilomes in plants. Molecular Plant 2022. 15 (1). 189-202)

4. Subsection 2.6. Please list the standards used.

5. How many basil leaves you took for analysis? Were sensory analysis repeated several times? Please add the appropriate information to MM Section.

6. Legend to Figure 1: Please add the number of picture before the name of basil species.

7. L 224: Kindly check the name of compounds “rosmatinic acid”.

Author Response

To editors and reviewers:

We really appreciated your hard work and valuable suggestions, here are our responses to your comments:

To reviewer #1:

  1. Abstract: I recommend adding the names of the nine studied basil species to the Abstract.

Response:

Revised accordingly.

  1. Introduction, L41: Kindly check the compound “rosinic acid”. It is not typical for basil.

Response:

In page 2, line 46, “rosinic acid” were revised to “linalool”.

  1. Introduction and references: Please check the reference # 18: the paper was published in 2022 (Honglun Yuan, Guangping Cao, Xiaodong Hou, Menglan Huang, Pengmeng Du, Tingting Tan, Youjin Zhang, Haihong Zhou, Xianqing Liu, Ling Liu, Yiding Jiangfang, Yufei Li, Zhenhuan Liu, Chuanying Fang, Liqing Zhao, Alisdair R. Fernie, Jie Luo. Development of a widely targeted volatilomics method for profiling volatilomes in plants. Molecular Plant 2022. 15 (1). 189-202)

Response:

In page 12, line 397, “Yuan, H.; Cao, G.; Hou, X.; Huang, M.; Du, P.; Tan, T.; Zhang, Y.; Zhou, H.; Liu, X.; Liu, L.; Jiangfang, Y.; Li, Y.; Liu, Z.; Fang, C.; Zhao, L.; Fernie, A. R.; Luo, J., Development of a widely targeted volatilomics method for profiling volatilomes in plants. Mol Plant 2021.” were revised to “Honglun Yuan, Guangping Cao, Xiaodong Hou, Menglan Huang, Pengmeng Du, Tingting Tan, Youjin Zhang, Haihong Zhou, Xianqing Liu, Ling Liu, Yiding Jiangfang, Yufei Li, Zhenhuan Liu, Chuanying Fang, Liqing Zhao, Alisdair R. Fernie, Jie Luo. Development of a widely targeted volatilomics method for profiling volatilomes in plants. Molecular Plant 2022. 15 (1). 189-202”.

  1. Subsection 2.6. Please list the standards used.

Response:

In page 4, line 170, we added “The standards used in the study are shown in supplementary table 1 (identification level A)”.

  1. How many basil leaves you took for analysis? Were sensory analysis repeated several times? Please add the appropriate information to MM Section.

Response:

In page 3, line 122, we added “To evaluate taste score, 0.5 g samples from each plant were prepared for nine panelists, all scores were averaged across panelists (n = 9) and plotted on a radar chart”.

  1. Legend to Figure 1: Please add the number of picture before the name of basil species.

Response:

Revised accordingly.

  1. L 224: Kindly check the name of compounds “rosmatinic acid”.

Response:

In page 7, line 233, “rosmatinic acid” were revised to “rosmarinic acid”.

Reviewer 2 Report

An interesting manuscript in which authors present an application of new methodology with higher accuracy for aromatic compounds determination,  applicable in pharmaceutical and food industry.

Author Response

To editors and reviewers:

We really appreciated your hard work and valuable suggestions, here are our responses to your comments:

To reviewer #2:

Response:

The manuscript were revised according to your suggestions.

Reviewer 3 Report

The research titled “Identification of key aromatic compounds in basil (Ocimum L.) using sensory evaluation, metabolomics and volatilomics analysis” is looks very promising on account of the idea and its presentation.  The needs and objectives of the project are very clear, and results have been presented up to the mark. The research team was able to identify nearly 180 volatiles and non-volatiles metabolites from the basil leaves which may have potential application in food industry, and for the crop genetics and breeding program to produce new cultivars with most promising aromatics. I have a few minor queries for the authors to be addressed prior considering the manuscript to be consider for publication.

Line 29-30: The conclusion statement in abstract should be more practical in terms of identifying the issues or the characteristics of the basil flavors which are required to be improved with upcoming research

Line 87 – 88: Authors can identify the perspective food industry where basil flavors may be used as an additive.

Line 92-93: Please provide the age of plant at the harvesting stage and that what part of plant was selected for leaf collection. If available, I would suggest authors to provide some details on plant characteristics. Basil aromatics have been reported to vary largely with the cultivar, age of the plant, plant height, leaf size,  and the environment. I would like to know if any of the factors listed above were kept uniform for all cultivars evaluated for metabolomics and volatilomics.  This information will certainly help researchers reproducing the results obtained.

Author Response

To editors and reviewers:

We really appreciated your hard work and valuable suggestions, here are our responses to your comments:

To reviewer #3:

  1. Line 29-30: The conclusion statement in abstract should be more practical in terms of identifying the issues or the characteristics of the basil flavors which are required to be improved with upcoming research

Response:

Revised accordingly.

  1. Line 87 – 88: Authors can identify the perspective food industry where basil flavors may be used as an additive.

Response:

In page 2, line 90, “We revealed the aromatic and metabolic differences within basil species and discovered the contributory compound to basil flavor, which provides a guide for its application in the food industry.” were revised to “We revealed the aromatic and metabolic differences within basil species and discovered the contributory compound to basil flavor, which provides a guide for its application in the food industry such as edible essence, pesto and so on”.

  1. Line 92-93: Please provide the age of plant at the harvesting stage and that what part of plant was selected for leaf collection. If available, I would suggest authors to provide some details on plant characteristics. Basil aromatics have been reported to vary largely with the cultivar, age of the plant, plant height, leaf size, and the environment. I would like to know if any of the factors listed above were kept uniform for all cultivars evaluated for metabolomics and volatilomics. This information will certainly help researchers reproducing the results obtained.

Response:

In page 3, line 97, we added “One to 1.5 cm width and 3-5 cm length leaves from eight months healthy plants (height around 40 to 60 cm) were harvested as samples. Leaf samples were collected randomly at the middle and upper parts from each plant, then cut into little pieces and divided into three replications. For much bigger leaves, leaf samples from No. 4 Ocimum gratissimum var. suave were collected with leaves of 5 cm width and 7-8 cm length”.
